# The Study of Bluetongue Virus (BTV) and Epizootic Hemorrhagic Disease Virus (EHDV) Circulation and Vectors at the Municipal Parks and Zoobotanical Foundation of Belo Horizonte, Minas Gerais, Brazil (FPMZB-BH)

**DOI:** 10.3390/v16020293

**Published:** 2024-02-15

**Authors:** Eduardo Alves Caixeta, Mariana Andrioli Pinheiro, Victoria Souza Lucchesi, Anna Gabriella Guimarães Oliveira, Grazielle Cossenzo Florentino Galinari, Herlandes Penha Tinoco, Carlyle Mendes Coelho, Zélia Inês Portela Lobato

**Affiliations:** 1Department of Preventive Veterinary Medicine (DMVP), Veterinary School, Universidade Federal de Minas Gerais (UFMG), Belo Horizonte 31270-901, Minas Gerais, Brazil; eduardoacaixeta@vetufmg.edu.br (E.A.C.); marianandrioli@gmail.com (M.A.P.); vslucchesi@gmail.com (V.S.L.); annagoliveira@gmail.com (A.G.G.O.); grazzicoss@gmail.com (G.C.F.G.); 2Belo Horizonte Municipal Parks and Zoobotany Foundation (FPMZB-BH), Belo Horizonte 31365-450, Minas Gerais, Brazil; herlandes@pbh.gov.br (H.P.T.); carlyle.m@pbh.gov.br (C.M.C.)

**Keywords:** *Culicoides* spp., zoo, deer diseases, hemorrhagic diseases

## Abstract

Bluetongue Virus (BTV) and Epizootic Hemorrhagic Disease Virus (EHDV) are *Orbiviruses* primarily transmitted by their biological vector, *Culicoides* spp. Latreille, 1809 (Diptera: Ceratopogonidae). These viruses can infect a diverse range of vertebrate hosts, leading to disease outbreaks in domestic and wild ruminants worldwide. This study, conducted at the Belo Horizonte Municipal Parks and Zoobotany Foundation (FPMZB-BH), Minas Gerais, Brazil, focused on *Orbivirus* and its vectors. Collections of *Culicoides* spp. were carried out at the FPMZB-BH from 9 December 2021 to 18 November 2022. A higher prevalence of these insects was observed during the summer months, especially in February. Factors such as elevated temperatures, high humidity, fecal accumulation, and proximity to large animals, like camels and elephants, were associated with increased *Culicoides* capture. Among the identified *Culicoides* spp. species, *Culicoides insignis* Lutz, 1913, constituted 75%, and *Culicoides pusillus* Lutz, 1913, 6% of the collected midges, both described as competent vectors for *Orbivirus* transmission. Additionally, a previously unreported species in Minas Gerais, *Culicoides debilipalpis* Lutz, 1913, was identified, also suspected of being a transmitter of these *Orbiviruses*. The feeding preferences of some *Culicoides* species were analyzed, revealing that *C. insignis* feeds on deer, Red deer (*Cervus elaphus*) and European fallow deer (*Dama dama*). Different *Culicoides* spp. were also identified feeding on humans, raising concerns about the potential transmission of arboviruses at the site. In parallel, 72 serum samples from 14 susceptible species, including various *Cervids*, collected between 2012 and 2022 from the FPMZB-BH serum bank, underwent Agar Gel Immunodiffusion (AGID) testing for BTV and EHDV. The results showed 75% seropositivity for BTV and 19% for EHDV. Post-testing analysis revealed variations in antibody presence against BTV in a tapir and a fallow deer and against EHDV in a gemsbok across different years. These studies confirm the presence of BTV and EHDV vectors, along with potential virus circulation in the zoo. Consequently, implementing control measures is essential to prevent susceptible species from becoming infected and developing clinical diseases.

## 1. Introduction

Bluetongue Virus (BTV) and Epizootic Hemorrhagic Disease Virus (EHDV) belong to the genus *Orbivirus* [1]. They possess a segmented double-stranded RNA (dsRNA) genome, an icosahedral capsid, and consist of ten genomic segments (seg-1 to seg-10), encoding seven structural proteins (VP1 to VP7) and four non-structural proteins (NS1, NS2, NS3/NS3a, and NS4). VP2 is identified as the primary immunodeterminant of the viral serotype [2,3,4]. These viruses, recognized for affecting a wide variety of vertebrate hosts, can trigger disease outbreaks in domestic and wild ruminants globally [5,6,7,8].

Currently, 27 notifiable serotypes of BTV are recognized by the World Organization for Animal Health (WOAH), adding the “atypical” serotypes, there are a total of 36 serotypes (BTV-1 to BTV-36) identified to date. In turn, EHDV has seven identified serotypes (EHDV-1, 2, 4, 5, 6, 7, and 8) [2,3,4,9]. The serotypes of both BTV and EHDV exhibit antigenic variation among themselves and can demonstrate different virulence, pathogenesis, tissue tropisms, and clinical signs in the same hosts. Animals that survive the infection develop specific antibodies against the infecting serotype. However, the absence of cross-protection increases the risk of outbreaks when introducing an exotic serotype to animals previously exposed to other virus serotypes [4,5,10,11,12].

BTV and EHDV are arboviruses primarily transmitted by their biological vector, *Culicoides* spp. Latreille, 1809 (Diptera: Ceratopogonidae), which become infected by feeding on the blood of viremic animals [13,14]. *Culicoides insignis* Lutz, 1913, and *C. pusillus* Lutz, 1913, are the only confirmed competent vectors in Brazil [13,14,15,16].

Infection with BTV and EHDV leads to systemic hemorrhages due to vascular injuries [17]. Clinical disease typically occurs when susceptible animals are introduced to enzootic areas or when new serotypes are introduced into endemic regions, highlighting the complex interplay among hosts, viral serotypes, vectors, and the environment [5,17]. Given the similarity in clinical signs and the potential for co-circulation of these viruses, laboratory diagnosis becomes essential to differentiate infections, utilizing both serological and molecular methods [11,12,18].

Both BTV and EHDV are classified as notifiable diseases by the World Organization for Animal Health [11,12] and the Brazilian Ministry of Agriculture, Livestock, and Supply (MAPA), in accordance with Normative Instruction No. 50, dated September 24, 2013 [19]. In Brazil, ten BTV serotypes (BTV-1, 3, 4, 12, 14, 17, 18, 19, 22, and 24) and two EHDV serotypes (EHDV-1 and 2) were identified through direct methods [20,21,22,23,24]. Furthermore, serum-positive samples for an additional nine BTV serotypes (BTV-2, 8, 9, 10, 13, 16, 20, 21, and 26) were reported [21,25,26]. Viral isolations of BTV were conducted in both, wild and domestic animals, providing more comprehensive information about this virus in South America compared to EHDV, which was isolated primarily from wild animals. Additionally, limited serological studies involving EHDV were conducted in this region.

Outbreaks of BTV and EHDV affecting *Cervids* have been documented in conservation centers, zoos, and wildlife establishments in Brazil. Despite the likely underreporting of cases, Brazil is considered an endemic region for BTV [20,24,25], and potentially for EHDV as well. Globally, BTV has caused and continues to cause outbreaks in various parts of the world. Notable incidents include the BTV-8 outbreaks that resulted in severe losses in Europe and the ongoing BTV-3 outbreak causing multiple cases across different European regions. The latter has already been identified in cattle and sheep, posing a risk of fatalities in various species and potential spread to other regions worldwide [5,7,27].

In 2020, a female Brow brocket (*Subulo gouazoubira*) succumbed to EHDV at the Belo Horizonte Municipal Parks and Zoobotany Foundation (FPMZB-BH), Minas Gerais, Brazil [28]. Our research focused on identifying various *Culicoides* spp. species inhabiting the FPMZB-BH from December 2021 to November 2022. This thorough investigation allowed us to delineate periods of heightened dipteran proliferation, pinpointing when the resident animals were most vulnerable to *Orbivirus* infections. Additionally, we delved into the origins of blood ingested by *Culicoides* spp. species to unveil potential hosts, aiming to elucidate the feeding preferences of these insects. Furthermore, we conducted a comprehensive assessment of sera stored in the FPMZB-BH serum bank from 2012 to 2022. Analyzing these archived samples over a decade facilitated the identification of susceptible species to these viruses and potential amplifying hosts. These studies have significantly contributed to the implementation of strategic control measures within the zoo and have advanced our comprehension of *Orbivirus* behavior in the Brazilian context.

## 2. Materials and Methods

### 2.1. Experiment Locations and Authorizations

The samples under study were sourced from the Belo Horizonte Municipal Parks and Zoobotany Foundation (FPMZB-BH), Minas Gerais, Brazil. Sample processing and subsequent laboratory tests took place at the Animal Virology Research Laboratory (LPVA) within the Department of Preventive Veterinary Medicine (DMVP) at the School of Veterinary Medicine, Federal University of Minas Gerais (UFMG).

Official authorization for this research was obtained from the Biodiversity Authorization and Information System (SISBIO) under reference number 80725. Ethical clearance was granted by the Ethics Committee on Animal Use (CEUA) at the Federal University of Minas Gerais (UFMG) with protocol number 233/2021. Furthermore, approval was secured from FPMZB-BH.

### 2.2. Culicoides Collection

The *Culicoides* spp. collections were conducted from 9 December 2021 to 18 November 2022, at the FPMZB-BH. A minimum of two collections per month were performed, every two weeks, on various dates, avoiding rainy days. In total, 25 collections were carried out, with each collection defined as the interval between the installation and removal of all traps. Trap placement occurred at 4:00 p.m., with retrieval at 8:00 a.m. the following day, following the guidelines proposed by Harrup [29].

Six traps were deployed for dipteran capture and subsequent species identification at FPMZB-BH. New Standard Miniature Blacklight traps, model 1212 (John W. Hock Company, Gainesville, FL, USA), suction-powered, 220 V, equipped with 8 W ultraviolet (UV) bulbs (model 1212), and one CDC Mini Light Trap with Incandescent Light modified to incorporate a UV bulb (John W. Hock Company, Gainesville, FL, USA), generously provided by the Pirbright Institute (Surrey, UK).

Each trap, labeled from 1 to 6 (A1 to A6), was strategically positioned as follows: A1 near the enclosures of the camel and llama, A2 and A3 near the enclosures of fallow deer, red deer, marsh deer, and oryx, A4 in a mixed enclosure housing tapirs, lowland paca, giant anteaters, and rheas, A5 near the enclosures of the waterbuck and zebra, and A6 between the enclosures of the elephants and white rhinoceros.

The selection of optimal points for trap installation involved assessing the proximity to hosts, choosing locations sheltered from rain, analyzing the surrounding vegetation, maintaining a recommended minimum distance between traps (at least 50 m), and considering nearby light sources that could interfere with captures. The traps were positioned at a minimum height of 1.5–2 m above the ground, following recommended protocols [29], and maintained the same arrangement throughout the entire experiment. 

Throughout all collections, data related to the date, sunrise and sunset times, maximum and minimum temperatures, maximum and minimum humidity, season, lunar brightness, rainfall, daily atmospheric pressure, presence of feces accumulation near the traps, trap performance during collections, temperature, and humidity at the installation and removal of each trap, as well as the total number of *Culicoides* spp. collected, were recorded. The collection period spanned from 9 December 2021 to 18 November 2022. The captured insects were stored in wide-mouth bottles, shielded from light, containing 70% alcohol, properly sealed, and labeled with information about the collection period and location in the zoo, then forwarded to the LPVA for species identification.

### 2.3. Identification of Culicoides spp.

The identification and characterization of *Culicoides* spp. species, including the determination of gender and gonadotropic stage for the analysis of the seasonal distribution of different species, were carried out following the procedures described by [29]. The characterization of wing pigmentation patterns for each specimen was performed using the identification keys from Farias [30], Castellón and Veras [31], Farias, Almeida and Pessoa [32], Santarém and Felippe-Bauer [15], Borkent and Dominiak [33], and Rios et al. [34].

### 2.4. Molecular Analysis of Culicoides spp. Blood Meal Hosts

Engorged females of species identified as potential vectors of BTV and EHDV were selected: *C. insignis*, *C. debilipalpis*, and *C. pusillus*. The engorged abdomens of *Culicoides* spp. were meticulously separated from the rest of the insect using entomological needles and processed following the protocol outlined by Carvalho et al. [35]. The identification of the species that the insects fed on was conducted using the cytochrome b (Cyt b) gene. All samples exhibiting a band on the 2% agarose gel for the Cyt b gene underwent sequencing at CT Vacinas (Minas Gerais, Brazil). Subsequently, a Basic Local Alignment Search Tool (BLAST^®^) search on the National Center for Biotechnology Information (NCBI) platform was performed for sample identification [36,37].

From these samples, it was possible to sequence and identify the blood meal sources of 12 individuals: 7 from *C. insignis*, 3 from C. debilipalpis, and 2 from *C. pusillus*.

### 2.5. Serum Samples from FPMZB-BH

A total of 72 serum samples, collected between 2012 and 2022 in accordance with zoo protocols and stored at −20 °C in the serum bank of FPMZB-BH, were tested. These samples originated from 62 animals representing 14 distinct species, including South American Tapir (*Tapirus terrestris*), Bactrian Camel (*Camelus bactrianus*), European Fallow Deer (*Dama dama*), Marsh Deer (*Blastocerus dichotomus*), Red Deer (*Cervus elaphus*), Waterbuck (*Kobus ellipsiprymnus*), Common Eland (*Taurotragus oryx*), African Bush Elephant (*Loxodonta africana*), Common Hippopotamus (*Hippopotamus amphibius*), Llama (*Lama glama*), Gemsbok (*Oryx gazella*), White Rhinoceros (*Ceratotherium simum*), Brown Brocket (*Subulo gouazoubira*), and Plains Zebra (*Equus quagga*).

The sera underwent testing for BTV and EHDV using Agar Gel Immunodiffusion (AGID) test. Antigens produced in the LPVA were employed, including EHDV-2 NEC2630 isolated in the LPVA [20] and BTV-4 provided by the Pan-American Foot-and-Mouth Disease Center. The testing procedures adhered to protocols outlined in the manuals provided by the WOAH [11,12].

### 2.6. Statistics

Geospatial and climatic data for the FPMZB-BH site (Pampulha, Belo Horizonte, Minas Gerais, Brazil) were collected using GPS devices. Weather information was sourced from The Weather Channel website (weather.com), accessed on all collection dates throughout the project. Additionally, data from the Brazilian National Institute of Meteorology (INMET) covering the entire period of *Culicoides* spp. captures were incorporated. These datasets were utilized to assess the situation regarding BTV and EHDV at the zoo.

Univariate and multivariate analyses were conducted to assess the relative risk of various *Culicoides* spp. collections, establishing relationships with factors such as air humidity, temperature, rainfall index, luminosity, season, wind speed, atmospheric pressure, and lunar phase. Statistical tools including Excel Windows, JAMOVI, and STATA were employed for these analyses.

## 3. Results

### 3.1. Collections and Identification of Culicoides spp.

A total of 568 specimens of *Culicoides* spp., representing various species, were captured. Detailed data for each capture can be found in Appendix A.

Among the traps, A1 had the highest capture (198 specimens), followed by A6 (108), and A2 had the lowest collection (36). For detailed data on the collection in each trap, refer to Appendix A. 

The most frequently identified species was *C. insignis*, totaling 426 specimens, followed by *C. paraensis* and *C. pusillus* with 35 and 32 specimens, respectively. Regarding the gonadotrophic stages, the most prevalent was the parous stage (298 specimens), followed by the nulliparous stage (156). Males and the gravid stage had the lowest counts, each totaling 27 specimens. Further detailed data, including the number of individuals captured for each species, group, gender, and gonadotrophic stage, are provided in Appendix A. 

In the summer months, characterized by higher temperatures and humidity, the peak *Culicoides* spp. capture occurred, reaching 241 specimens in February. A decline was observed during the colder and drier winter months, with no individuals found in September. Refer to Appendix A for monthly data on the quantity of specimens captured, broken down by species. The total number of individuals captured each month can be visualized in Appendix A. 

Univariate and multivariate analyses, assessing the relative risk of *Culicoides* spp. captures (Appendix A), indicated that temperature, humidity, precipitation, trap location, the presence of feces accumulation near traps, and the date of collection were the most influential factors. It is noteworthy that these variables may exhibit associations, complicating independent comparisons. For instance, temperature and humidity strongly correlate with the collection date, and the presence of feces accumulation near traps correlates with trap location. These interrelationships among factors may have influenced the analyses.

It is noteworthy that, in addition to the *Culicoides* species previously recorded in the state of Minas Gerais, *C. debilipalpis* Lutz, 1913, was identified for the first time in this state.

### 3.2. Molecular Analysis of Culicoides spp. Blood Meal Hosts

The results of the sequenced engorged blood samples for each *Culicoides* spp. species can be viewed in Table 1.

Belo Horizonte Zoos do not have any of the following species identified in Table 1: chital (*Axis axis*), sambar deer (*Cervus unicolor*), Red collared dove (*Streptopelia tranquebarica*), brown creeper (*Certhia americana*), Iranian lizard (*Agamura kermansis*), or Amazonian fish (*Teleocichla mindanensis*). These results could are probably due to an incomplete availability of sequences from the conserved region of *Cyt b* in databases for some possible hosts, especially Brazilian species, and due to the presence of highly similar sequences. 

### 3.3. Serological Tests

The results of the AGID tests for BTV and EHDV are available in Table 2.

A South American Tapir showed seropositivity for BTV in a sample collected in 2017. However, in a sample from 2019, it was found to be seronegative, and in a sample collected in 2020, it once again exhibited seropositivity. Regarding a European Fallow Deer, it was seronegative for BTV in the 2013 sample but showed seropositivity for this virus in the 2014 sample. On the other hand, a Gemsbok, which initially tested seronegative for EHDV in a 2012 sample, turned out to be seropositive for this virus in the 2014 sample. These results show that these viruses are circulating in the zoo.

## 4. Discussion

Following the death of a female Brow brocket (*Subulo gouazoubira*) due to EHDV at FPMZB-BH in 2020 [28], this research was formulated to investigate the presence of *Orbivirus* vectors and evaluate the seropositivity for BTV and EHDV of the animals inhabiting this environment. The primary objective was to comprehend the circulation of these viruses within the zoo context and to discern their potential implications for the overall health of the resident animals. 

*C. insignis* emerged as the predominant species in our collections, consistently observed across all traps and in most months, except during the winter period from June to September. This observation aligns with a study conducted by Laender et al. [38], which identified *C. insignis* as the most prevalent species in Minas Gerais. Carvalho and Silva’s research [39] in Maranhão, Brazil, also emphasized the prevalence of *C. insignis*, particularly in locations with mammals, such as corrals and pigsties.

The second most commonly found species was *C. pusillus*, raising concerns as this species, along with *C. insignis*, is identified as a vector for BTV and EHDV [13,16]. 

This is the first identification of *C. debilipalpis* in the state of Minas Gerais, Brazil, despite its presence in various other regions of South and Southeast Brazil [33]. *C. debilipalpis*, previously described as a potential vector of BTV and EHDV in North America [40,41,42,43,44], is known to feed on *Orbivirus* hosts, and viral replication of BTV and EHDV in this insect was confirmed when feeding on infected blood [42]. According to the literature reviews conducted, this study is believed to be the first to document the identification of *C. debilipalpis* feeding on African elephants and tapirs, both potential *Orbivirus* hosts. Notably, in contrast to previous studies describing the preference of *C. debilipalpis* for *Cervids*, no individuals in the present study fed on species from this group [40,45,46]. Consequently, further studies are imperative to confirm the role of *C. debilipalpis* in the transmission cycle of BTV and EHDV.

The distribution of the collected samples, analyzed in terms of gender and gonadotropic stage, aligns with previous research [47,48] where parous females were observed in greater or equal quantities compared to other stages.

A generalist feeding behavior of *C. insignis* was identified, supporting previous studies where this species was observed feeding on birds, rodents, and dogs [49]. It is plausible that the studied species exhibit opportunistic behavior, adapting their feeding habits to nearby species, regardless of the host’s nature. That can be a possibility for the identification that all *Culicoides* spp. species feed in human samples in the blood meal analysis (*Homo sapiens*). This implies a higher possibility of arbovirus transmission [50] and the bite of *C. insignis* is associated with allergic dermatitis in different species [51]. This behavior may be attributed to the migratory pattern of vertebrates and the small size of *Culicoides* spp., which can be carried several kilometers by the wind [52]. This adaptation would be essential for these insects to meet their nutritional needs in different environments and locations [35,47,50]. These could be some of the reasons for the broad range of hosts of *Culicoides* spp. Studies suggest that generalist species may experience a reduction in their vectorial capacity due to the wide variety of hosts they feed on. However, this generalist feeding habit may allow for the transmission of pathogens across a broader range of hosts, contributing to the maintenance of cycles of different diseases [50,53]. Additionally, generalist species tend to be more abundant as they can survive in diverse environments, facilitating transmission and assisting in the maintenance of cycles of different diseases [35]. An important finding was the substantial abundance of *C. insignis* sequences originating from Red Deer and Fallow Deer, suggesting a potential preference for feeding on *Cervids* by this *Culicoides* species.

Despite the complexity of inferring feeding preferences based on the analysis of blood ingested by *Culicoides* spp., this approach provides valuable insights into which species deserve greater attention, particularly in the case of potential vectors of BTV and EHDV, such as *C. insignis*, *C. pusillus*, and C. *debilipalpis*. Understanding the different hosts of each *Culicoides* spp. species is crucial for the development of more accurate predictive models regarding pathogen transmission and the formulation of effective prevention strategies for various diseases [54]. 

In the hot and humid months, characteristic of summer, such as February, more *Culicoides* spp. individuals and species were collected compared to the cold and dry months, such as September. In the winter collections, a reduction in the capture of all species was observed, with some not being found for several months, such as *C. pusillus* and *C. foxi*. This is due to the fact that *Culicoides* spp. are thermophilic species, and this characteristic, combined with the reduction of the extrinsic incubation period (EIP) due to higher temperatures in the summer, points to a higher risk of *Orbivirus* infection in hot and humid months [13,16]. Precipitation also influences the selection of days for trap placement, resulting in a greater number of insects captured on rain-free days, despite humidity being important to this dipteran reproduction. *Culicoides* spp., being small insects, are adversely affected by heavy rains, particularly accompanied by wind, which interferes with the collections [16,55]. 

In a detailed analysis of how the location of each trap affected the collections, we identified that the presence of an accumulation of feces near traps and proximity to water sources proved to be relevant factors. That is due to the fact that *Culicoides* spp. feed and reproduce in humid places with the presence of organic matter [16]. Another identified aspect of increased midge capture is related to placing traps outside enclosures, aligning with the exophagic and exophilic behavior of these insects [56]. 

Another factor that may have influenced the number of insects collected could be the presence of certain nearby hosts. It is possible that a higher number of insects collected near camels (Trap A1) and elephants (Trap A6) could be due to the larger amount of feces produced by these animals, less competition for space for these hematophagous insects, and less activity of larger animals. Also, previous research has identified that the proximity of camels to traps could lead to an increase in collections in those areas [47,57]. 

The combination of these variables resulted in variations in the collection rates of *Culicoides* spp., as well as in the number of species collected in each trap. Trap A1, which showed the highest collection, encompassed all the key factors identified in the analysis. Notably, the presence of *C. insignis* was more pronounced in this trap compared to other species, suggesting the existence of some specific attractant for this particular species.

When comparing FPMZB-BH with other zoos worldwide that have been the subject of *Culicoides* spp. collection projects, we observe a great diversity of collected species, similar to the present study. However, several previous studies have reported higher quantities of *Culicoides* spp. collected compared to the current project. For instance, at the National Zoological Gardens of South Africa, between 2002 and 2004, Labuschagne et al. [48] collected 478,040 specimens, and in another study at Chester Zoo in 2008, Vilar et al. [57] collected 35,401 specimens. However, a study conducted by Nelder et al. [58] at Greenville Zoo and Riverbanks Zoo in 2007 collected 101 and 88 *Culicoides* spp., respectively, numbers closer to those obtained in the present research. This variation can be attributed to the different environmental conditions of each zoo and the distinct habits of *Culicoides* spp. species in each region.

A higher prevalence of seropositive animals for BTV was observed, totaling 75%, compared to EHDV, which recorded 19%. This disparity suggests a potentially more significant circulation of BTV in the FPMZB-BH region, although the specific serotypes are still unknown. However, the considerable number of seronegative animals for EHDV raises concerns about the susceptibility of these animals to potential infection by this virus.

It is crucial to highlight that the majority of *Cervids* from different species exhibited seropositivity for BTV, with several also testing positive for EHDV. The identification of *C. insignis* feeding on *Cervids* indicates that these species are at risk of infection.

African ruminants (Gemsboks, Waterbuck, and Eland), Camelids (llamas and camels), and the tested elephant showed seropositivity for *Orbiviruses*, corroborating previous findings [10,47,59,60,61], suggesting the possible involvement of these African ruminants in these viruses’ cycle. 

The seropositivity of a tapir for BTV, as revealed in this study, aligns with findings, in free-ranging animals, from a prior study conducted in the Brazilian savanna by Fernandes-Santos et al. [62], which also identified seropositive individuals of this species for BTV. It is noteworthy that tapirs are listed as a vulnerable species and are distributed throughout Brazil [63]. Moreover, our data show variation in serological results over time in an animal of this species, suggesting the possibility of a decline in antibody levels, reinfection, or exposure to new viral serotypes. Associated with these findings the identification of *C. debilipalpis* feeding on tapir emphasizes the potential involvement of these animals in the epidemiology of BTV and EHDV in Brazil. 

The observation of seroconversion in animals from one year to another, both for EHDV (in the case of one gemsbok) and BTV (one fallow deer and one tapir), suggests that these animals were infected within the zoo. Therefore, it is recommended to conduct RT-PCR tests for viremia detection in animals before new animals are introduced in the zoo and in cases of transferring animals between zoos. 

## 5. Conclusions

This study identified *Orbivirus* vectors and seropositive animals in the zoo, confirming the circulation of BTV and EHDV in the FPMZB-BH, which are infecting different mammal species.

## Figures and Tables

**Table 1 viruses-16-00293-t001:** Results of the identified sequences of Cyt b from blood meal of different *Culicoides* spp.

*Culicoides* spp.	Identified Host Species
*Culicoides insignis*	Red deer (*Cervus elaphus*), European Fallow Deer (*Dama dama*), Chital (*Axis axis*), *Sambar deer* (*Cervus unicolor*), Human (*Homo sapiens*), Red Collared Dove (*Streptopelia tranquebarica*), Iranian lizard (*Agamura kermansis*), Amazonian fish (*Teleocichla mindanensis*), Brown creeper (*Certhia americana*)
*Culicoides debilipalpis*	African bush elephant (*Loxodonta Africana*), Human (*Homo sapiens*), South American Tapir (*Tapirus terrestris*)
*Culicoides pusillus*	Human (*Homo sapiens*)

**Table 2 viruses-16-00293-t002:** Results of the AGID test for BTV and EHDV by species using serum samples collected in FPMZB-BH from 2012 to 2023.

Species		BTV AGID Results			EHDV AGID Results		
Common Name	Scientific Name	Positive	Negative	Total	Positive	Negative	Total
African Bush Elephant	*Loxodonta africana*	1	0	1	0	1	1
Bactrian Camel	*Camelus bactrianus*	2	3	5	0	5	5
Brown Brocket	*Subulo gouazoubira*	9	4	13	4	7	11
Common Eland	*Taurotragus oryx*	1	0	1	0	1	1
Common Hippopotamus	*Hippopotamus amphibiu*	0	1	1	0	1	1
European Fallow Deer	*Dama dama*	17	1	18	2	11	13
Gemsbok	*Oryx gazella*	9	0	9	3	3	6
Llama	*Lama glama*	5	1	6	1	5	6
Marsh Deer	*Blastocerus dichotomus*	1	0	1	0	1	1
Plains Zebra	*Equus quagga*	0	2	2	0	2	2
Red Deer	*Cervus elaphus*	6	0	6	1	4	5
South American Tapir	*Tapirus terrestris*	2	3	5	0	5	5
Waterbuck	*Kobus ellipsiprymnus*	1	1	2	0	1	1
White Rhinoceros	*Ceratotherium simum*	0	2	2	0	1	1
Total		54	18	72	11	48	59 *

* Thirteen samples (five European Fallow Deer, one Red Deer, one Waterbuck, three Gemsbok, one White Rhinoceros, and two Brown Brocket) were not tested for EHDV due to insufficient volume.

## Data Availability

All data are presented in the main manuscript.

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
