# Peer review of "The Study of Bluetongue Virus (BTV) and Epizootic Hemorrhagic Disease Virus (EHDV) Circulation and Vectors at the Municipal Parks and Zoobotanical Foundation of Belo Horizonte, Minas Gerais, Brazil (FPMZB-BH)"

_viruses, 2024, doi:10.3390/v16020293_

Round 1

Reviewer 1 Report

Comments and Suggestions for Authors

The study written by "Caixeta et al." and titled "Study of Bluetongue Virus (BTV) and Epizootic Hemorrhagic Disease Virus (EHDV) circulation and vectors at the Municipal Parks and Zoobotanical Foundation of Belo Horizonte, Minas Gerais, Brazil (FPMZB-BH)" describes the variety of species of arthropod vectors captured in a zoo in Brazil, evaluating the species on which these hosts take blood meals, and evaluating the serological positivity to EHDV and BTV of susceptible animals present in the zoo. Overall, the manuscript presents sufficient data for publication, is written in an acceptable manner, and provides conclusions based on the results obtained. However, the manuscript requires revision to make it easier to read (some sections need to be summarized). Below are my comments.

Abstract: The abstract is quite long, although information about the vectors and the virus is missing (it starts directly with materials and methods). Some sentences (e.g., Lines 20–23) can be summarized without lists of results. This way, it is smoother and easier to understand.

Lines 14–15: "where a clinical case of orbiviruses occurred in a female brow brocket (Subulo gouazoubira) that died from an EHDV infection in 2020." This information, repeated in the manuscript, must be accompanied by a bibliographic reference. In any case, it is not the reason why it is important to carry out surveillance and monitoring of infections transmitted by culicoides.

Line 80: I suggest the authors also briefly talk about the (rapidly expanding) situation of BTV and EHDV in pets. These viruses are also widespread in Europe, as evidenced by studies conducted in the Campania region of Turkey (BTV), China, Italy, and Portugal. I believe this information is important to make the reader understand the uncontrolled spread of these viruses among both domestic and wild animals.

Line 100: More information on sample storage and how they were sampled (for sera) is needed.

The authors should verify the necessity of all references (materials and methods already over 50, perhaps a little too many).

Subsections 2.5, 2.6, and 2.7 could be included in only one subsection.

Line 172: It is difficult to consider “an epidemiological study” the survey made by authors with 72 samples.

Figure 1: It would be appropriate to include this image in supplementary materials.

Results Subsections 2.2 and 2.3: These subsections are quite limited. Various information present in the discussion can be moved here.

Discussion: Authors should eliminate references to tables (they are part of the results) from the discussion and limit themselves to comparing their results with those of other similar studies by discussing the implications of their research.

Comments on the Quality of English Language

The level of English is good

Author Response

Good afternoon,

Thank you very much for reviewing the article "Study of Bluetongue Virus (BTV) and Epizootic Hemorrhagic Disease Virus (EHDV) circulation and vectors at the Municipal Parks and Zoobotanical Foundation of Belo Horizonte, Minas Gerais, Brazil (FPMZB-BH)."

Your comments were very helpful in improving the article.

Regarding the abstract, the results list was removed, and the most relevant information was added in subsequent sentences. Additionally, the information about the EHDV infection case in 2020 in the brow brocket (Subulo gouazoubira) at the zoo has not been published in a journal yet. However, I described this case in my master's thesis, and I will use it as the suggested reference. Also, I removed this information from the introduction since it is not the main reason for the study.

Information about the samples was moved from line 100 and was more detailed in subsection 2.5.

Section 2.7 was removed as the limited number of samples makes it complex to conduct an epidemiological study.

Figure 1 was transformed into a table for easier readability.

Additional information was added in subsections 3.1, 3.2, and 3.3 of the results, as requested.

Some references to the tables were removed from the discussion.

Information about the global expansion of Orbiviruses and the associated risks was added, as suggested.

The necessity of all references was reviewed, and some references were removed.

Thank you again for the review, and attached is the new version of the article.

Best regards,

Eduardo Alves Caixeta

Reviewer 2 Report

Comments and Suggestions for Authors

Dear Authors

Bluetongue (BT) and epizootic hemorrhagic disease (EHD) cases have increased worldwide, causing significant economic loss to ruminant livestock production and detrimental effects to susceptible wildlife populations. 

The paper is original and contains certain novelties.

The summary sufficiently informs about the content of the paper.

Abstract includes introductory statement that outlines the background and significance of the study.

Introduction summarizes relevant research to provide context and clearly state the problem.  The topics are well developed and confronted to other publications.

The discussion section interprets the findings in view of the results obtained in this and in past studies on this topic.

Minor corrections:

Line 30, whose samples were collected? “Also, serum samples collected from 2012 to 2022 were received. A total of 72 serums from 14  30 species were tested and of these,” Rephrase this sentence so that it is clear. What species? Include some information as shown in 158-166.

They describe BTV much more than EHDV. More information about EHDV should also be provided.

Materials and methods should be complete enough to allow possible replication of the research. Some information is missing about Agar Gel Immunodiffusion (AGID).

Remove sp. and spp. in italics throughout the manuscript.

Uniformity is lacking in the references.

Author Response

Good afternoon,

Thank you very much for reviewing the article "Study of Bluetongue Virus (BTV) and Epizootic Hemorrhagic Disease Virus (EHDV) circulation and vectors at the Municipal Parks and Zoobotanical Foundation of Belo Horizonte, Minas Gerais, Brazil (FPMZB-BH)." Your comments were very helpful in improving the article.

Regarding the suggested corrections:

The sentence in line 30 has been reformulated for better understanding of the samples.

Additional information about the study of EHDV in South America has been included, as requested.

The description of the Agar Gel Immunodiffusion (AGID) test has been expanded to facilitate the replication of the experiment.

The italicized "sp." and "spp." have been removed.

References have been standardized.

Thank you again for the review, and attached is the new version of the article.

Best regards,

Eduardo Alves Caixeta

Round 2

Reviewer 1 Report

Comments and Suggestions for Authors

The authors have addressed many of my comments, greatly improving the manuscript. The item is almost ready for acceptance. I recommend the authors convert Tables 1, 2, and 3 and Figure 1 into supplementary files (they describe the sampling rather than the results).

Author Response

Good afternoon,

The tables and figures in question have been transferred to the supplementary material as requested. Some of the results present in these tables have been described in section 3.1 of the results. Additionally, the enclosures near each trap have been added to the description of Culicoides collection in the methodology, section 2.2.

Thank you very much for the suggestions!

Best regards,

Eduardo Alves Caixeta
